# Learning Conformal Explainers for Image Classifiers

## Abstract

Feature attribution methods are widely used for explaining image-based predictions, as they provide feature-level insights that can be intuitively visualized. However, such explanations often vary in their robustness and may fail to faithfully reflect the reasoning of the underlying black-box model. To address these limitations, we propose a novel conformal prediction–based approach that enables users to directly control the fidelity of the generated explanations. The method identifies a subset of salient features that is sufficient to preserve the model's prediction, regardless of the information carried by the excluded features, and without demanding access to ground-truth explanations for calibration. Four conformity functions are proposed to quantify the extent to which explanations conform to the model's predictions. The approach is empirically evaluated using five explainers across six image datasets. The empirical results demonstrate that FastSHAP consistently outperforms the competing methods in terms of both fidelity and informational efficiency, the latter measured by the size of the explanation regions. Furthermore, the results reveal that conformity measures based on super-pixels are more effective than their pixel-wise counterparts.

## 1 Introduction

Many of the state-of-the-art machine learning algorithms operate as black boxes, which not only limits the user's ability to understand the rationale behind their decisions but also constrains their adoption in high-stakes domains. The transparency is necessary not only for establishing trust in predictive models but also for addressing compliance with legal and regulatory requirements, as well as ethical obligations (Goodman & Flaxman, 2017). Therefore, explainable machine learning has emerged as a prominent research area that achieves interpretability while maintaining high predictive performance.

Explainable machine learning methods employ a set of strategies to make model behavior more understandable. Among the most common strategies are the construction of counterfactual examples (Karimi et al., 2020; Mothilal et al., 2020; Van Looveren & Klaise, 2021; Guo et al., 2021), the use of local interpretable surrogate models (Ribeiro et al., 2016; Lundberg & Lee, 2017), the identification of important features (Chen et al., 2018; Yoon et al., 2019; Jethani et al., 2021), and feature attribution methods (Simonyan et al., 2014; Sundararajan et al., 2017; Lundberg & Lee, 2017; Schwab & Karlen, 2019; Covert & Lee, 2021; Jethani et al., 2022).

Feature attribution methods are particularly prominent since they provide detailed information by assigning an importance score to each input feature, which can provide fine-grained insights into which specific inputs mainly drive the model's decision. Moreover, the attributed importance scores lend themselves to intuitive visualizations, e.g., heatmaps for image data or bar plots for tabular inputs. However, feature attributions are not always robust (Hsieh et al., 2021; Fel et al., 2023), and their fidelity to the underlying black-box model may vary (Yeh et al., 2019; Lakkaraju & Bastani, 2020), i.e., the extent to which an explanation accurately captures the decision-making behavior of the underlying model. The low fidelity can have adverse consequences, especially in human–AI hybrid decision-making settings (Sadeghi et al., 2024; Cabitza et al., 2024). Generally, explanations are provided without associated uncertainty estimates, and methods that do offer post-hoc uncertainty quantification, e.g., (Alkhatib et al., 2023) and (Alkhatib et al., 2024), rely on access to ground-truth explanations for offline calibration, which may not always be feasible, particularly in the case of

image data. Furthermore, such approaches preclude the user from controlling the fidelity of the generated explanations.

Therefore, the main contributions of this work are:

- **a novel method based on conformal prediction** that enables users to directly control the fidelity of the generated explanations while also providing an indication of their uncertainty
- **the method determines sufficient explanations** that can reproduce the predictions of the underlying black-box model at a user-specified confidence level
- **a set of conformity functions** specifically designed to quantify the degree to which the explanations conform to the model's predictions

The next section introduces fundamental concepts of the conformal prediction framework and feature attribution explanations, along the way, we also introduce the notation used throughout the paper. Section 3 details the proposed method, while Section 4 outlines the experimental setup and presents the results of the empirical investigation. Section 6 provides a brief overview of related work. Section 5 reports the time complexity of the proposed method. Finally, the concluding remarks summarize the main findings and discuss potential directions for future research.

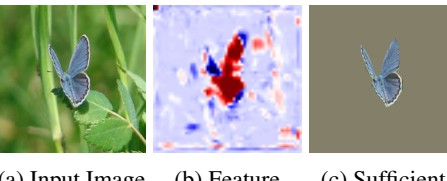

(a) Input Image    (b) Feature Attribution    (c) Sufficient Explanation

Figure 1: An example of the output explanation (subfigure c) using the proposed method with 85% confidence.

## 2    PRELIMINARIES

In this section, we provide a concise overview of the key concepts underlying the proposed approach. We begin by introducing conformal prediction in the following subsection, and subsequently present explanation methods based on feature attribution, with particular emphasis on the Shapley value as a representative example.

### 2.1    CONFORMAL PREDICTION

Conformal prediction has emerged as a prominent approach for uncertainty estimation within the machine learning community. Originally introduced as a transductive approach, conformal prediction required training a separate model for each test instance, making it computationally expensive (Gammerman et al., 1998; Saunders et al., 1999). Consequently, the inductive conformal prediction has been proposed by Papadopoulos et al. (2002), which trains a single model on provided data and employs it for predictions on new instances. For simplicity, we refer to inductive conformal prediction as conformal prediction throughout the remainder of this manuscript. Conformal prediction methods construct prediction sets that include the true target with a predefined probability (a property referred to as validity), employing past observations to determine precise levels of confidence in new predictions (Shafer & Vovk, 2008). Under the assumption that a given calibration dataset $\mathbb{Z}$ consists of $k$ independent and identically distributed (i.i.d.) pairs of inputs and labels $\mathbb{Z} = \{(\boldsymbol{x}_1, y_1), (\boldsymbol{x}_2, y_2), \ldots, (\boldsymbol{x}_k, y_k)\}$, a conformal predictor assigns a p-value to a candidate pair $(\boldsymbol{x}_{k+1}, y_{k+1})$, where $y_{k+1}$ is a potential label for a new observation $\boldsymbol{x}_{k+1}$ drawn from the same distribution. The definition of p-value here is intertwined with the non-conformity measure ($\alpha_i$), which quantifies the degree of 'strangeness' of an example, i.e., how unusual a data point appears relative to observed data examples (Toccaceli & Gammerman, 2017). There is no universal function for measuring non-conformity. Nevertheless, simple choices for the non-conformity measure are often sufficient. For instance, in regression problems, one may use the absolute error ($\alpha_i = |y_i - \tilde{y}_i|$) (Papadopoulos et al., 2002), where $\tilde{y}_i$ is the predicted outcome by the underlying model; or ($\alpha_i = 1 - P_f(y_i \mid x_i)$) in classification, where $P_f(y_i \mid x_i)$ denotes the assigned probability to label $y_i$ by model $f$. Therefore, the p-value corresponding to a candidate label $y_{k+1} \in \mathbb{Y}$, where $\mathbb{Y}$ is the set of possible labels for each prediction, is defined by (Vovk, 2012):

$$p^{y_{k+1}} = \frac{|\{(\boldsymbol{x}_i, y_i) \in \mathbb{Z} : \alpha_i \geq \alpha_{y_{k+1}}\}| + 1}{k + 1}.$$

Given a predefined significance level $\epsilon \in (0,1)$ and a sequence of non-conformity scores $\mathbb{A} = \{\alpha_1, \alpha_2, \ldots, \alpha_k\}$, the smallest $\alpha_\epsilon \in \mathbb{A}$ such that:

$$\frac{|\{(\boldsymbol{x}_i, y_i) \in \mathbb{Z} : \alpha_i \leq \alpha_\epsilon\}| + 1}{k+1} \geq 1 - \epsilon, \tag{1}$$

can be employed to construct a prediction set as follows:

$$\{y_{k+1} \in \mathbb{Y} : \alpha_{y_{k+1}} \leq \alpha_\epsilon\}.$$

The specified non-conformity score ($\alpha_\epsilon$) in equation 1 guarantees that the resulting prediction sets contain the true label with a probability mass that meets or exceeds the specified confidence level $1 - \epsilon$.

*Regardless of the non-conformity measure employed, the claim that the prediction sets contain the true label with confidence level $1 - \epsilon$ holds under the assumption that the data are i.i.d. (Shafer & Vovk, 2008). The claim also holds under the slightly relaxed assumption that the data samples are probabilistically exchangeable (Vovk et al., 2005). However, the choice of the non-conformity function impacts the efficiency of the conformal predictor (Linusson et al., 2020; Aleksandrova & Chertov, 2021).*

## 2.2 EXPLANATIONS VIA FEATURE ATTRIBUTION

The proposed method can be applied to explanation techniques that produce feature attributions. Explaining model predictions through feature attribution is a class of explanation methods that assign a score to each input feature, reflecting the feature's proportional contribution to the model's output. A prominent example of feature attribution methods is the Shapley value, which can be used to construct an additive explanation model $\mu$ that serves as an interpretable approximation of a value function $v$. The value function quantifies how the model's prediction changes when different subsets of features are marginalized out. The explanation model $\mu$ can be expressed as follows (Lundberg & Lee, 2017):

$$\mu(N) = \phi_0(v) + \sum_{i \in N} \phi_i(v),$$

where $\phi_0(v)$ is a constant and $\phi_i(v)$ the marginal contribution of feature $i \in N = \{1, 2, \ldots, n\}$. $\mu(N)$ can be learned by minimizing the following weighted least squares loss function for a given instance $\boldsymbol{x} \in \boldsymbol{X}$ (Covert & Lee, 2021):

$$\mathcal{L}(v_x, \mu_x) = \sum_{S \subseteq N} \omega(S) \Big( v_x(S) - \mu_x(S) \Big)^2, \tag{2}$$

with $\omega$ a weighting kernel and $S \subseteq N$. Since the proposed approach involves selecting subsets of features, it is important to highlight standard approaches to remove features (an essential consideration in Shapley-value-based explanation methods). There are three commonly used value functions $v$ for marginalizing out features that are not included in a coalition $S$:

1. Interventional approach (Marginal Expectations/Random Baseline) (Chen et al., 2020): $v_x(S) = \mathbb{E}_{\boldsymbol{x}_S} \big[ h\big(\boldsymbol{x}_S, \boldsymbol{X}_{N \setminus S}; \theta\big) \big]$, where $h$ is a predictive model. This approach replaces the removed features with independent random values drawn from the marginal distribution, i.e., breaks dependence between selected and masked features, which may lead to unrealistic input combinations off the data manifold.

2. Observational approach (Conditional Expectations) (Chen et al., 2020; Zern et al., 2023): $v_x(S) = \mathbb{E}_{\boldsymbol{x}_S} \big[ h\big(\boldsymbol{X}_S; \theta\big) | \boldsymbol{X}_S = \boldsymbol{x}_S \big]$. Here, the masked features are sampled conditionally based on the selected subset $\boldsymbol{x}_S$, therefore, preserving dependencies present in the data, and the generated combinations remain in distribution.

3. Baseline removal approach (Sundararajan & Najmi, 2020): $v_x(S) = h\left(\boldsymbol{x}_S, \tilde{\boldsymbol{x}}_{N \setminus S}; \theta\right)$, where $\tilde{\boldsymbol{x}}$ is a fixed baseline vector that often takes the values of $\mathbb{E}\left[\boldsymbol{X}\right]$

We adopt the baseline removal approach due to its simplicity of implementation and compatibility with a broad range of feature attribution methods that do not rely on Shapley value estimation, e.g., CXPlain (Schwab & Karlen, 2019) and Integrated Gradients (Sundararajan et al., 2017), where the baseline can be conveniently set to a zero vector, particularly when standard scaling is applied and feature values are centered around zero.

The fidelity of the attributions to the underlying black-box model can vary both across different explanation methods and across individual data instances (Yeh et al., 2019; Hsieh et al., 2021; Fel et al., 2023; Wang et al., 2024; Miró-Nicolau et al., 2024). Additionally, such attributions do not precisely determine the exact subset of features responsible for the prediction of the black-box model. The proposed method employs conformal prediction to identify the minimal subset of important input features, based on the attributed scores, that is sufficient to preserve the model's prediction while ensuring that the explanation approximation error remains within a prespecified significance level.

## 3 THE PROPOSED METHOD

We propose a method that not only enables users to quantify the uncertainty associated with explanations but also provides direct control over the fidelity of the generated explanations.

In what follows, we adopt the notion of a conformity measure, which, while similar to the concept of non-conformity, serves a complementary purpose, i.e., rather than quantifying the strangeness of an example, the conformity measure quantifies how typical or common it appears in comparison to observed examples. *Our objective is to compute a conformity function that accurately identifies the smallest subset of features sufficient for the model to produce the same prediction as it would using all features for a given data point.* The proposed method operates on post-hoc explanations and grants users control over the acceptable error rate, defined as the discrepancy between the model's original prediction and the prediction based on the identified important regions of the input features.

### 3.1 ALGORITHM

We define a function $\psi(\boldsymbol{x}, \Phi; \sigma_\epsilon)$ that post-processes the explanations of a model $h$, where $\psi$ is parameterized by $\sigma_\epsilon$ that controls the level of conservativeness, and $\Phi$ represents the set of attributed scores for the input features of $\boldsymbol{x}$. Lower values of $\sigma_\epsilon$ lead to more conservative explanations, i.e., larger sets of features are identified as important to maintain the prediction. The output of $\psi$ is a subset of features that contains the necessary features for model $h$ to yield its prediction for a given test instance $\boldsymbol{x}$, with the probability of omitting an essential feature upper bounded by $\epsilon$.

The parameter $\sigma_\epsilon$ is learned using a held-out calibration set $\mathbb{Z} = \{(\boldsymbol{x}_1, \Phi_1), (\boldsymbol{x}_2, \Phi_2), \ldots, (\boldsymbol{x}_k, \Phi_k)\}$ over which we compute conformity measures $\sigma_i, \forall \boldsymbol{x}_i \in \mathbb{Z}$. The conformity measures quantify how typical a feature is as part of a sufficient explanation relative to previously observed explanation examples. More specifically, for each data instance $\boldsymbol{x}_i \in \mathbb{Z}$, the conformity function identifies the highest threshold $\sigma_i$ such that the prediction made using only the subset $S_i$ of features with all attribution scores $\phi_f^{(i)} \geq \sigma_i$ matches the prediction obtained when using the grand coalition of all features. Formally, let $\Phi = \{\phi_1, \phi_2, \ldots, \phi_d\}$ denote the attribution scores assigned to the input features $\boldsymbol{x} = \{f_1, f_2, \ldots, f_d\}$, we define conformity measure $\sigma_i$ for instance $\boldsymbol{x}_i$ as:

$$\sigma_i = \sup\left\{\tau \in \Phi_i : S_{i,\tau} = \left\{f_j \in \boldsymbol{x}_i : \phi_j^{(i)} \geq \tau\right\}, \arg\max h(S_{i,\tau}) = \arg\max h(\boldsymbol{x}_i)\right\}, \quad (3)$$

where $\tau$ can take values from quantile levels over the range of $\Phi_i$ values, or alternatively, from the discrete set of attribution scores in $\Phi_i$. For a significance level $\epsilon \in (0, 1)$ and a set of conformity scores $\boldsymbol{\sigma} = \{\sigma_1, \sigma_2, \ldots, \sigma_k\}$, we define $\sigma_\epsilon$ as follows:

$$\sigma_\epsilon = \sup\left\{\sigma \in \boldsymbol{\sigma} : \frac{|\{(\boldsymbol{x}_i, \Phi_i) \in \mathbb{Z} : \sigma_i \geq \sigma\}| + 1}{k+1} \geq 1 - \epsilon\right\}.$$

Finally, for new data examples $\boldsymbol{x}_{k+1}$, $\psi$ returns a sufficient explanation set $S_E^{(k+1)}$ that is constructed as follows:

$$S_E^{(k+1)} = \left\{f_j \in \boldsymbol{x}_{k+1} : \phi_j^{(k+1)} \geq \sigma_\epsilon\right\}.$$

Consequently, a feature $j$ is included in the sufficient explanation coalition $S_E^{(k+1)}$ if its attribution score satisfies $\phi_j \geq \sigma_\epsilon$, in which case its corresponding value $f_j$ is retained. Otherwise, the feature is omitted from $S_E^{(k+1)}$, and its value remains at its baseline value in $\tilde{\boldsymbol{x}}$.

**Theorem 1** *Assume $\mathbb{Z} \cup (\boldsymbol{x}_{k+1}, \Phi_{k+1})$ are i.i.d., the probability of excluding a feature from $S_E^{(k+1)}$ that is important for preserving the model's predicted class, $\arg\max h(\boldsymbol{x}_{k+1})$, is upper bounded by the predefined significance level $\epsilon$.*

The proof can be found in Appendix B.

The explanation returned by $\psi$ is guaranteed to include the important features, sufficient to maintain the prediction, with probability at least $1 - \epsilon$. Nevertheless, a certain explainer tends to produce concise and informative explanation sets $S_E^{(k+1)}$, whereas an explainer that assigns importance scores at random may result in excessively large sets, potentially encompassing the entire input.

An obvious limitation of explanations based on individual important pixels is that they often result in sparse and, possibly, noisy representations both for the user and the predictive model. Consequently, we also propose the subsequent conformity measures.

## 3.2 SUPER-PIXEL-BASED CONFORMITY FUNCTION

While the default conformity function used in the proposed algorithm (as defined in equation 3) determines the explanation set $S_E^{(k+1)}$, it may result in highly fragmented image regions. Therefore, we introduce an alternative conformity function that incorporates the image's segmentation structure, thereby enabling $\psi$ to return a set of superpixels as a coherent and sufficient explanation.

The super-pixel-based conformity function applies any standard image segmentation algorithm to compute superpixels $\gamma_l \subset \boldsymbol{x}_i$ and defines $\sigma_i^{(sp)}$ as the largest attribution threshold such that the model's prediction, restricted to the subset $S_i$ of superpixels that contain at least a proportion $\rho$ of features satisfying $\phi_j^{(i)} \geq \sigma_i^{(sp)}$ $\left(\text{i.e., } |\{\phi_j^{(i)} \in \gamma_l : \phi_j^{(i)} \geq \sigma_i^{(sp)}\}| \geq \rho\right)$, matches the prediction obtained using the grand coalition of all features. $\sigma_i^{(sp)}$ is defined as follows:

$$\sigma_i^{(sp)} = \sup\left\{\tau \in \Phi_i : S_{i,\tau} = \left\{\gamma_l \subset \boldsymbol{x}_i : |\{\phi_j^{(i)} \in \gamma_l : \phi_j^{(i)} \geq \tau\}| \geq \rho\right\}, \arg\max h(S_{i,\tau}) = \arg\max h(\boldsymbol{x}_i)\right\},$$

(4)

where $\tau$ as mentioned before, can take values from the attribution scores within $\Phi_i$ or can be the quantile levels over the range of $\Phi_i$, and the sufficient explanation set $S_E^{(k+1)}$ for a new data example $\boldsymbol{x}_{k+1}$ is formed as follows:

$$S_E^{(k+1)} = \left\{\gamma_l \subset \boldsymbol{x}_{k+1} : |\{\phi_j^{(k+1)} \in \gamma_l : \phi_j^{(k+1)} \geq \sigma_\epsilon^{(sp)}\}| \geq \rho\right\}.$$

Theorem 1 also extends to the super-pixel–based conformity function, where the super-pixels are considered as features rather than individual pixels.

### 3.3 SCALED ATTRIBUTION SCORES (SCALED VALUES)

This function operates similarly to equation 4, but the attribution scores of each data instance are standardized by centering them to a zero mean and scaling them to unit variance. Each explanation $\Phi_i$ is normalized by subtracting its mean ($\bar{\varphi}_i$) and dividing by its standard deviation ($std_i$) as shown below:

$$\Phi_i^{scaled} = \frac{\Phi_i - \bar{\varphi}_i}{std_i}$$

### 3.4 SUMMATION BASED CONFORMITY FUNCTION (SUMMED VALUES)

Instead of returning a threshold for each data instance, the summation-based conformity function ($\sigma_i^\Sigma$) identifies the smallest subset of features, ranked by their attribution scores, whose cumulative sum is minimal and sufficient to preserve the model's original prediction. $\sigma_i^\Sigma$ is formally defined in equation 5.

$$\sigma_i^\Sigma = \inf\left\{ \sum_{f_j \in S_{i,\tau}} |\phi_j^{(i)}| \ : \ S_{i,\tau} = \left\{ f_j \in \boldsymbol{x}_i \ : \ \phi_j^{(i)} \geq \tau \right\}, \ \arg\max h(S_{i,\tau}) = \arg\max h(\boldsymbol{x}_i) \right\}. \quad (5)$$

Then $\sigma_\epsilon^\Sigma$ is defined as follows:

$$\sigma_\epsilon^\Sigma = \inf\left\{ \sigma^\Sigma \in \boldsymbol{\sigma} \ : \ \frac{|\{(\boldsymbol{x}_i, \Phi_i) \in \mathbb{Z} : \sigma_i^\Sigma \leq \sigma^\Sigma\}| + 1}{k + 1} \geq 1 - \epsilon \right\},$$

and for a sufficient explanation set $S_E^{(k+1)}$ for a new data instance $\boldsymbol{x}_{k+1}$ let $\pi_{k+1} = [f_1, f_2, \ldots, f_d]$ be the list of features ordered such that $\phi_1 \geq \phi_2 \geq \ldots \geq \phi_d$. Then, the sufficient explanation set is:

$$S_E^{(k+1)} = \left\{ f_j \in \pi_{k+1} : \left( \sum_{r=1}^{j} \phi_r \right) \leq \sigma_\epsilon^\Sigma \right\}.$$

## 4 EMPIRICAL INVESTIGATION

We assess the proposed approach using 4 conformity functions across 6 distinct image datasets and 5 different explainers. The evaluation compares both the conformity functions and the explainers in terms of their informativeness, where producing more concise explanation sets corresponds to higher informativeness, and the fidelity of the determined explanations, i.e., the degree to which the explanations preserve the same prediction of the model when provided with the full input image.

### 4.1 EXPERIMENTAL SETUP

Each dataset is partitioned into training, validation, and evaluation subsets. The training set is used to fit the predictive black-box models, the validation set serves to monitor overfitting and determine early stopping, and the evaluation set is reserved for assessing model performance. The evaluation set is further divided into a calibration subset and a test subset, where the calibration subset is employed to compute the conformity score at a predefined significance level, while the test subset is used to evaluate both the proposed conformity functions and the employed explainers. For the feasibility of the experiments, $\tau$ in Equations (3), (4), and (5) takes values from the quantile levels over the range of $\Phi_i$. Superpixel segmentation is performed using the SLIC algorithm (Achanta et al., 2012). All images are normalized with mean and standard deviation of the training set. The black-box models are based on ResNet50 (He et al., 2016). The employed explainers are Saliency (Simonyan et al., 2014), InputXGradient (Shrikumar et al., 2016), KernelSHAP, GradientSHAP (Lundberg &

Lee, 2017), and FastSHAP (Jethani et al., 2022). A detailed description of the employed datasets is provided in Appendix H.

The results are evaluated based on the informativeness of the resulting explanation sets, adopting a paradigm analogous to the evaluation of the informational efficiency of conformal predictors, where efficiency is measured by the size of the prediction regions and sets, i.e., smaller prediction regions are more informative (Linusson et al., 2020; Messoudi et al., 2020; Alkhatib et al., 2023). For example, Figure 2 presents the explanations generated by 2 competing explainers for a prediction of an English Springer image taken from the Imagenette dataset. Adjacent to each explanation, we display its corresponding extracted $S_E$ region obtained using our proposed approach. The extracted $S_E$ regions in subfigures (c) and (e) successfully reproduce the original prediction of the English Springer. However, the $S_E$ region derived using FastSHAP in subfigure (c) is smaller, thereby demonstrating superior informational efficiency. In the following experiments, the fidelity quantifies the empirical agreement between the model's prediction on the sufficient explanation ($S_E$) and the full input. Formally, fidelity is computed as the ratio between the number of test instances for which both predictions coincide and the total number of test instances.

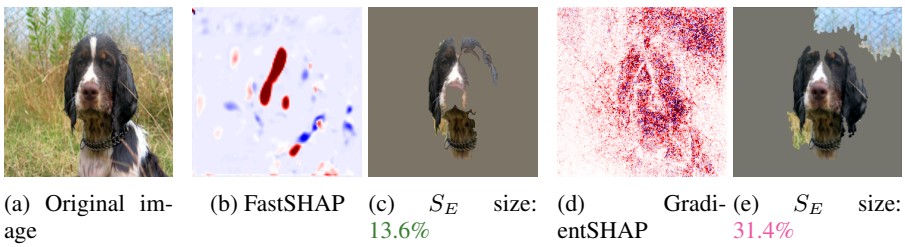

(a) Original image    (b) FastSHAP    (c)   $S_E$   size: 13.6%    (d)   GradientSHAP    (e)   $S_E$   size: 31.4%

Figure 2: Explanations generated by FastSHAP and GradientSHAP for an English Springer image from the Imagenette dataset, along with their corresponding extracted $S_E$ regions at 95% confidence obtained using the Scaled Values conformity function. The shown $S_E$ regions successfully reproduce the original prediction as an English Springer.

## 4.2 Informational Efficiency Evaluation

We evaluate the informational efficiency of the conformal explainers using several baseline approaches (KernelSHAP, FastSHAP, GradientSHAP, Saliency, and InputXGradient), while also evaluating the efficiency of the proposed conformity measures in decisively distinguishing between conforming and non-conforming patterns. Figure 3 shows the average size of the extracted explanation regions ($S_E$) at different confidence levels for each explainer using the Scaled Values function. The detailed results on the six datasets using the four conformity functions at 95% confidence level are available in Table 1 in Appendix C. The results indicate that FastSHAP is generally a more efficient explainer than the competing approaches, while KernelSHAP shows high uncertainty and produces substantially larger $S_E$ regions, therefore, its explanations are comparatively less informative.

The comparison of the proposed conformity functions, as illustrated in Figure 4, indicates that the super-pixel-based functions (Super-Pixels and Scaled Values) result in more informationally efficient explanations than the pixel-based functions (Pixelwise and Summed Values) at high confidence levels. Particularly, by aggregating information over coherent regions rather than evaluating individual pixels in isolation, the super-pixel–based measures can capture higher-level structural patterns, which leads to explanations that are more compact and less noisy. In contrast, pixel-based measures tend to produce fragmented and less efficient explanations, as the importance values can be distributed across many fine-grained elements, as illustrated in Figure 5. Moreover, applying scaling to the obtained attribution values (via the Scaled Values function) serves to regulate their magnitudes, thereby reducing the size of the determined explanation regions at higher confidence levels, as detailed in Table 1 and Table 3.

## 4.3 Fidelity Evaluation

The confidence levels allow the user to explicitly control the trade-off between compactness and reliability of the explanations. As shown in Figure 3, FastSHAP achieves high fidelity levels while

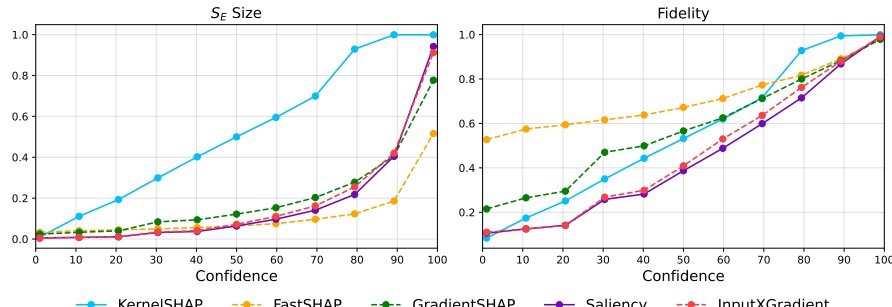

Figure 3: The effect of varying confidence levels on the size of $S_E$ and the corresponding fidelity levels, obtained using the Scaled Values conformity function on the Animal-10 dataset.

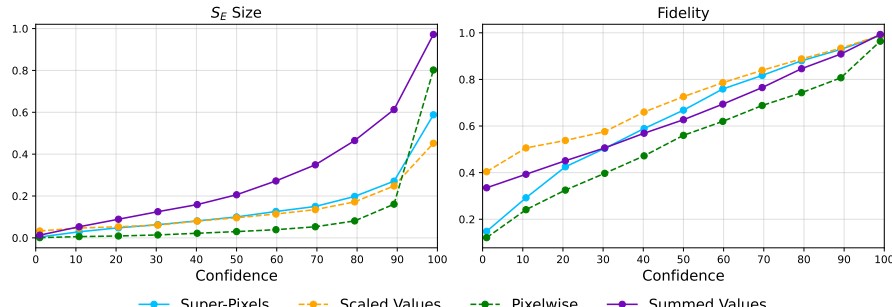

Figure 4: The effect of the confidence level on the size of $S_E$ and the fidelity level of the retrieved explanations using FastSHAP with the proposed conformity functions on Imagenette dataset.

simultaneously maintaining relatively small explanation regions. Additionally, Figure 4 illustrates that the Scaled Values function offers comparatively higher fidelity levels while maintaining relatively small $S_E$ sizes. In contrast, the Summed Values function is the least efficient and results in the largest $S_E$ sizes.

The comparison of explainers in terms of fidelity, as shown in Table 2, reveals comparable accuracy in reproducing the black-box model's predictions across the same conformity function. However, super-pixel–based functions produce fidelity levels that are more consistent with the predefined confidence level than pixel-wise functions. Therefore, the super-pixel–based functions are shown not only to be more decisive in separating conforming from non-conforming patterns but also more reliable in maintaining the validity guarantees. Table 2 presents the detailed fidelity evaluation results for the compared explainers across the four proposed conformity functions, and Table 4 presents the fidelity results of FastSHAP at different confidence levels. The results are also summarized for all the datasets and the conformity functions in Figure 7. Figure 6 provides an illustrative example from the Imagenette dataset, demonstrating how increasing the confidence level, and thereby enforcing stricter fidelity guarantees, affects the resulting explanations.

## 5 TIME COMPLEXITY

We report the computational complexity of the function in equation 4, which is the most efficient (with scaling) and the most computationally demanding function. First, the complexity of segmenting all images is $\mathcal{O}(N \times M)$, as we employ the SLIC algorithm (Achanta et al., 2012).

**Calibration Complexity:** The computational cost of the calibration procedure is primarily determined by repeated model evaluations across multiple importance thresholds. For a calibration set of $N$ images of size $M = H \times W$, a model evaluation cost $\Omega$, and $P$ percentile thresholds, the overall complexity can be expressed as: $\mathcal{O}\big(P \times (N \times M + N \times \Omega)\big)$. Here, the first term accounts for

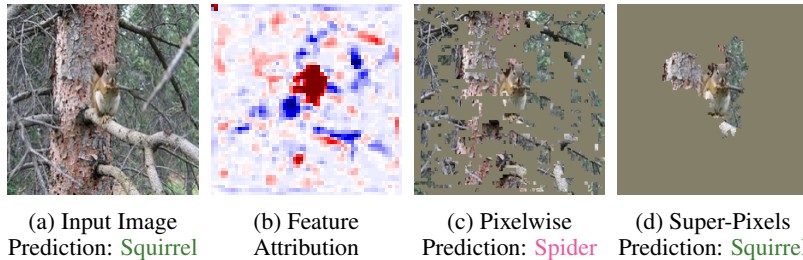

|  |  |  |  |
|---|---|---|---|
| (a) Input Image
Prediction: Squirrel | (b) Feature
Attribution | (c) Pixelwise
Prediction: Spider | (d) Super-Pixels
Prediction: Squirrel |

Figure 5: The sufficient explanation ($S_E$) determined using the pixel-wise conformity function (sub-figure c) and the super-pixel–based function with scaled values (subfigure d). The example is obtained from the Animal-10 dataset at 95% confidence.

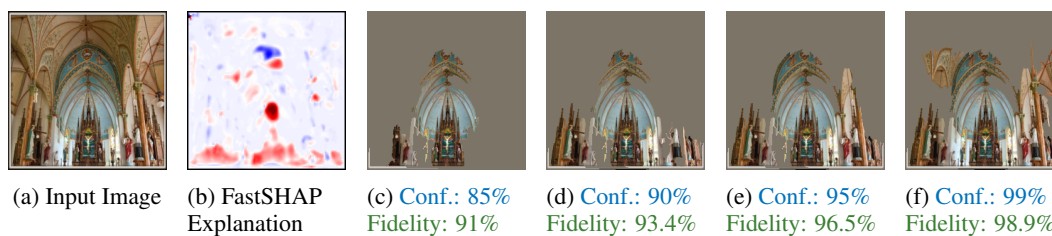

|  |  |  |  |  |  |
|---|---|---|---|---|---|
| (a) Input Image | (b) FastSHAP
Explanation | (c) Conf.: 85%
Fidelity: 91% | (d) Conf.: 90%
Fidelity: 93.4% | (e) Conf.: 95%
Fidelity: 96.5% | (f) Conf.: 99%
Fidelity: 98.9% |

Figure 6: The effect of varying confidence levels on the trade-off between the fidelity and compactness of explanations. Fidelity is reported on the test set for each confidence level. The example is obtained from the Imagenette dataset.

superpixel extraction, mask generation, and thresholding operations, each linear in the number of pixels. The second term corresponds to repeated forward passes of the black-box model on masked images. Since model inference typically dominates the computational cost, the total complexity can be approximated as: $\mathcal{O}(P \times N \times \Omega)$.

**Inference Complexity:** At inference time, the computational complexity is primarily linear in the number of pixels per image. Specifically, the procedure involves: Computing pixel-level importance masks via thresholding with $\sigma_\epsilon$, with complexity $\mathcal{O}(M)$. Aggregating pixel-wise importance over superpixels and applying a filtering criterion, also $\mathcal{O}(M)$. Element-wise masking of the images, which incurs $\mathcal{O}(C \times M)$, where $C$ denotes the number of channels in the input. Consequently, the overall inference complexity is: $\mathcal{O}(C \times M)$. In practice, the computational cost is dominated by simple array operations and scales linearly with image size, rendering the procedure highly efficient.

## 6 RELATED WORK

The conformal prediction framework has increasingly been employed as a means to quantify the uncertainty associated with explanations, as it provides distribution-free and finite-sample valid guarantees. Alkhatib et al. (2023) and Idrissi et al. (2025) combined the conformal prediction framework with feature attributions to derive feature-wise uncertainty estimates, thereby quantifying the reliability of individual attribution scores. In contrast, Alkhatib et al. (2024) employed conformal prediction to generate uncertainty indicators at the level of the entire explanation encompassing all the attributed scores. The Venn prediction framework (Vovk et al., 2003; Lambrou et al., 2015), a statistical framework closely related to conformal prediction for quantifying uncertainty, has also been explored for calibrating feature attributions (Löfström et al., 2024; Löfström et al., 2024). The impact of employing conformal prediction and Venn prediction to calibrate the predictions of the underlying black-box model, and consequently the generated explanations, has been investigated by Löfström et al. (2023) and Mehdiyev et al. (2025a). Moreover, the conformal prediction framework has been applied to provide fidelity guarantees for explanatory rules extracted from regression models (Johansson et al., 2022), while Alkhatib et al. (2022) proposed quantifying the uncertainty of post-hoc explanatory rules using the Venn prediction framework. Nevertheless, the mentioned

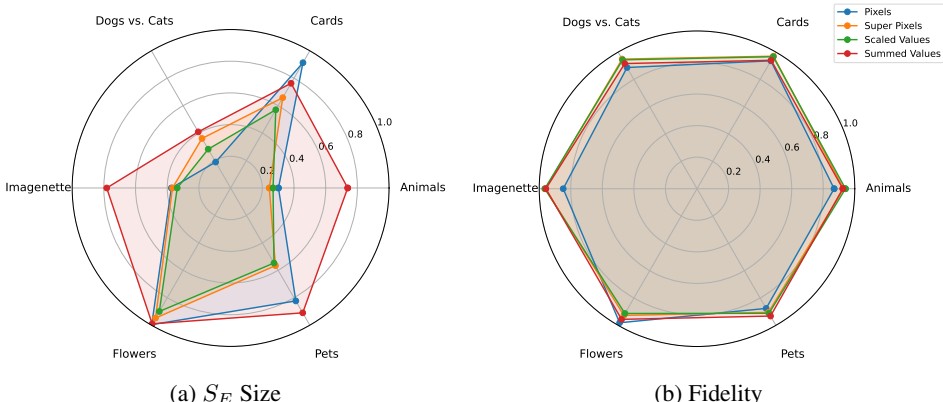

(a) $S_E$ Size                          (b) Fidelity

Figure 7: The average size and fidelity of the explanation regions determined at 95% confidence using FastSHAP. In subfigure (a), the Scaled Values function generally produces the most efficient, whereas in subfigure (b), all conformity functions achieve comparable levels of fidelity.

approaches have been developed primarily for tabular data, and their applicability to image datasets remains limited.

Beyond the use of the conformal prediction framework for uncertainty quantification of explanations, research at the intersection of interpretable machine learning and conformal prediction has explored a variety of approaches. Prior work has either combined the predictions of interpretable models with conformal prediction or employed conformal prediction to enhance the interpretability of the underlying model (Johansson et al., 2019a;b; Martinez Gil et al., 2024; Narteni et al., 2026). In addition, Jaramillo & Smirnov (2021) proposed employing Shapley values as conformity scores for inductive conformal predictors instead of the common conformity scores, highlighting the potential of integrating attribution methods with conformal prediction.

Apart from conformal prediction, alternative approaches for quantifying the uncertainty of the explanations have been proposed. Schulz et al. (2022) investigated quantifying the uncertainties of surrogate model explanations. Hill et al. (2024) proposed to integrate uncertainty due to the complexity of the decision boundary with the uncertainty that arises from the approximation of the explanation function. For image classification, Zhang et al. (2022) developed an uncertainty quantification–based framework to interpret deep neural network decisions. Additionally, Mehdiyev et al. (2025b) combined predictive uncertainty with post-hoc explanations to quantify and explain the uncertainty in the predictions of a machine learning model. However, such approaches are typically not model-agnostic, lack the statistical guarantees offered by the conformal prediction framework, and do not enable users to explicitly control the fidelity level of the generated explanations.

## 7 CONCLUDING REMARKS

We have introduced a novel algorithm based on conformal prediction that determines explanation regions that are sufficient to preserve the predictions of the underlying black-box model. The proposed algorithm enables the user to balance the fidelity and the informativeness of the determined explanations. Moreover, the proposed algorithm does not require ground truth values for calibration. To this end, we proposed four conformity functions designed to estimate the extent to which the attributed importance scores conform to the model's predictions. Our empirical evaluation, conducted across 6 image datasets, assessed both the informational efficiency and the fidelity of the resulting explanations. The results reveal that conformity functions based on super-pixels consistently outperform pixel-based conformity functions in terms of both efficiency and fidelity. Additionally, the results indicate that FastSHAP outperforms the competing explainers with respect to the efficiency of their explanations. Future research directions include the design of explanation algorithms that explicitly incorporate uncertainty information, enabling the construction of more robust and reliable explanations, drawing inspiration from the conformal training of Stutz et al. (2022). Another promising direction is the training of real-time explainers, e.g., FastSHAP, using super-pixels derived.

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

## A  LIMITATIONS

The proposed approach requires calibration data and the generation of explanations for the black-box model's predictions on the provided data, which can be computationally intensive for some explainers, e.g., KernelSHAP, which trains a surrogate white-box model separately for each prediction. Consequently, the method may be less suitable in scenarios with limited data availability. At inference time, however, the approach remains efficient, since constructing the sufficient explanation $S_E$ requires only a single score ($\sigma_\epsilon$) at the specified confidence level. Hence, the main computational burden remains in computing $\sigma_\epsilon$ offline.

## B  PROOF OF THEOREM 1

Let us define the random variable

$$\lambda = \frac{|\{(\boldsymbol{x}_i, \Phi_i) \in \mathbb{Z} : \sigma_i \geq \sigma_{k+1}\}| + 1}{k + 1}.$$

Under the i.i.d. assumption of $\mathbb{Z} \cup (\boldsymbol{x}_{k+1}, \Phi_{k+1})$, using the results from Proposition 1 of Papadopoulos et al. (2002) it holds that

$$\mathbb{P}(\lambda \leq \epsilon) \leq \epsilon,$$

where $\mathbb{P}$ is a probability measure. Therefore,

$$\mathbb{P}(\sigma_{k+1} \leq \sigma_\epsilon) \leq \epsilon.$$

Let $j \in \boldsymbol{x}_{k+1}$ be a feature whose omission results in a prediction change. Then,

$$j \notin S_E^{(k+1)} \iff \phi_j^{(k+1)} < \sigma_\epsilon \iff \sigma_{k+1} < \sigma_\epsilon.$$

Thus,

$$\mathbb{P}(j \notin S_E^{(k+1)}) \leq \mathbb{P}(\sigma_{k+1} \leq \sigma_\epsilon) \leq \epsilon.$$

## C INFORMATIONAL EFFICIENCY EVALUATION

The study evaluates the efficiency of conformal explainers with (KernelSHAP, FastSHAP, GradientSHAP, Saliency, and InputXGradient) as well as the efficiency of the proposed conformity measures. Results show that FastSHAP generally produces the most efficient explanations. Among conformity functions, super-pixel–based measures produce more compact explanations compared to pixel-based measures, which are fragmented and less efficient. Additionally, scaling attribution values further reduces explanation size by regulating their magnitude.

Table 1: The informational efficiency of the five explainers, KernelSHAP, FastSHAP, GradientSHAP, Saliency, and InputXGradient, at 95% confidence level using proposed conformity measures.

| Conformity Function | Dataset | KernelSHAP | FastSHAP | GradientSHAP | Saliency | InputXGradient |
|---|---|---|---|---|---|---|
| Super-Pixels | Animals 10 | $0.945 \pm 0.228$ | $\mathbf{0.244 \pm 0.1}$ | $0.675 \pm 0.246$ | $0.687 \pm 0.241$ | $0.659 \pm 0.238$ |
| | Cards | $0.989 \pm 0.106$ | $\mathbf{0.658 \pm 0.126}$ | $0.935 \pm 0.099$ | $0.932 \pm 0.096$ | $0.938 \pm 0.094$ |
| | Dogs and Cats | $0.955 \pm 0.207$ | $\mathbf{0.361 \pm 0.134}$ | $0.49 \pm 0.244$ | $0.502 \pm 0.268$ | $0.461 \pm 0.241$ |
| | Imagenette | $0.955 \pm 0.208$ | $\mathbf{0.367 \pm 0.181}$ | $0.641 \pm 0.257$ | $0.589 \pm 0.264$ | $0.561 \pm 0.257$ |
| | Flowers 102 | $0.957 \pm 0.202$ | $0.948 \pm 0.122$ | $0.822 \pm 0.187$ | $0.824 \pm 0.184$ | $\mathbf{0.79 \pm 0.178}$ |
| | Oxford IIIT Pet | $0.959 \pm 0.197$ | $\mathbf{0.566 \pm 0.146}$ | $0.681 \pm 0.246$ | $0.67 \pm 0.255$ | $0.673 \pm 0.231$ |
| Scaled Values | Animals 10 | $0.944 \pm 0.23$ | $\mathbf{0.268 \pm 0.123}$ | $0.592 \pm 0.149$ | $0.583 \pm 0.14$ | $0.583 \pm 0.152$ |
| | Cards | $0.928 \pm 0.258$ | $\mathbf{0.57 \pm 0.155}$ | $0.904 \pm 0.099$ | $0.953 \pm 0.059$ | $0.955 \pm 0.067$ |
| | Dogs and Cats | $0.957 \pm 0.202$ | $\mathbf{0.283 \pm 0.11}$ | $0.308 \pm 0.093$ | $0.305 \pm 0.09$ | $0.33 \pm 0.101$ |
| | Imagenette | $0.95 \pm 0.219$ | $\mathbf{0.337 \pm 0.168}$ | $0.384 \pm 0.125$ | $0.479 \pm 0.145$ | $0.499 \pm 0.156$ |
| | Flowers 102 | $0.957 \pm 0.203$ | $0.9 \pm 0.189$ | $0.873 \pm 0.105$ | $0.837 \pm 0.098$ | $\mathbf{0.833 \pm 0.116}$ |
| | Oxford IIIT Pet | $0.953 \pm 0.211$ | $\mathbf{0.547 \pm 0.145}$ | $0.678 \pm 0.148$ | $0.689 \pm 0.146$ | $0.724 \pm 0.148$ |
| Pixelwise | Animals 10 | $0.948 \pm 0.222$ | $\mathbf{0.302 \pm 0.079}$ | $0.953 \pm 0.019$ | $0.824 \pm 0.088$ | $0.957 \pm 0.02$ |
| | Cards | $0.992 \pm 0.087$ | $0.913 \pm 0.029$ | $0.97 \pm 0.014$ | $0.858 \pm 0.115$ | $0.977 \pm 0.01$ |
| | Dogs and Cats | $0.957 \pm 0.204$ | $\mathbf{0.189 \pm 0.079}$ | $0.884 \pm 0.042$ | $0.793 \pm 0.085$ | $0.899 \pm 0.044$ |
| | Imagenette | $0.958 \pm 0.202$ | $\mathbf{0.374 \pm 0.148}$ | $0.795 \pm 0.082$ | $0.674 \pm 0.08$ | $0.81 \pm 0.09$ |
| | Flowers 102 | $0.977 \pm 0.149$ | $0.996 \pm 0.006$ | $0.992 \pm 0.011$ | $\mathbf{0.98 \pm 0.021}$ | $0.99 \pm 0.01$ |
| | Oxford IIIT Pet | $0.989 \pm 0.104$ | $0.823 \pm 0.083$ | $0.982 \pm 0.019$ | $0.821 \pm 0.146$ | $0.981 \pm 0.021$ |
| Summed Values | Animals 10 | $0.984 \pm 0.031$ | $\mathbf{0.739 \pm 0.39}$ | $0.949 \pm 0.162$ | $0.926 \pm 0.199$ | $0.942 \pm 0.177$ |
| | Cards | $0.97 \pm 0.056$ | $0.763 \pm 0.361$ | $0.979 \pm 0.091$ | $0.913 \pm 0.224$ | $0.979 \pm 0.088$ |
| | Dogs and Cats | $0.996 \pm 0.004$ | $\mathbf{0.409 \pm 0.419}$ | $0.945 \pm 0.172$ | $0.946 \pm 0.151$ | $0.95 \pm 0.166$ |
| | Imagenette | $0.975 \pm 0.043$ | $\mathbf{0.782 \pm 0.384}$ | $0.865 \pm 0.273$ | $0.877 \pm 0.249$ | $0.907 \pm 0.235$ |
| | Flowers 102 | $0.993 \pm 0.002$ | $0.99 \pm 0.0$ | $0.99 \pm 0.0$ | $0.99 \pm 0.0$ | $0.99 \pm 0.0$ |
| | Oxford IIIT Pet | $0.991 \pm 0.016$ | $\mathbf{0.91 \pm 0.244}$ | $0.99 \pm 0.0$ | $0.945 \pm 0.156$ | $0.99 \pm 0.0$ |

# D  THE FIDELITY OF THE DETERMINED EXPLANATIONS

The fidelity comparison shows that while all explainers achieve comparable accuracy in reproducing model predictions, super-pixel–based conformity functions provide more consistent fidelity with the confidence level than pixel-wise functions. Therefore, the super-pixel–based conformity functions are more decisive in distinguishing conforming vs. non-conforming explanations and more reliable in preserving validity guarantees than the pixel-wise conformity functions.

Table 2: The fidelity of the explanations obtained using the five explainers, KernelSHAP, FastSHAP, GradientSHAP, Saliency, and InputXGradient, at 95% confidence level using proposed conformity measures.

| Conformity | Dataset | KernelSHAP | FastSHAP | Grad.SHAP | Saliency | InputXGrad. |
|---|---|---|---|---|---|---|
| Super-Pixels | Animals 10 | 0.957 | 0.93 | 0.954 | 0.956 | 0.946 |
| | Cards | 0.989 | 0.985 | 0.97 | 0.936 | 0.857 |
| | Dogs and Cats | 1 | 0.949 | 0.95 | 0.963 | 0.947 |
| | Imagenette | 0.957 | 0.96 | 0.97 | 0.961 | 0.96 |
| | Flowers 102 | 0.958 | 0.927 | 0.914 | 0.917 | 0.913 |
| | Oxford IIIT Pet | 0.964 | 0.905 | 0.924 | 0.939 | 0.937 |
| Scaled Values | Animals 10 | 0.948 | 0.941 | 0.946 | 0.936 | 0.932 |
| | Cards | 0.928 | 0.977 | 0.966 | 0.909 | 0.845 |
| | Dogs and Cats | 0.958 | 0.943 | 0.901 | 0.913 | 0.902 |
| | Imagenette | 0.955 | 0.965 | 0.94 | 0.946 | 0.938 |
| | Flowers 102 | 0.957 | 0.913 | 0.914 | 0.9 | 0.902 |
| | Oxford IIIT Pet | 0.955 | 0.912 | 0.91 | 0.918 | 0.921 |
| Pixelwise | Animals 10 | 0.958 | 0.869 | 0.905 | 0.899 | 0.907 |
| | Cards | 0.989 | 0.989 | 0.936 | 0.811 | 0.736 |
| | Dogs and Cats | 1 | 0.887 | 0.787 | 0.898 | 0.806 |
| | Imagenette | 0.96 | 0.848 | 0.902 | 0.872 | 0.904 |
| | Flowers 102 | 0.978 | 0.981 | 0.811 | 0.845 | 0.788 |
| | Oxford IIIT Pet | 0.99 | 0.876 | 0.846 | 0.88 | 0.848 |
| Summed Values | Animals 10 | 0.973 | 0.924 | 0.932 | 0.926 | 0.924 |
| | Cards | 0.992 | 0.974 | 0.94 | 0.902 | 0.838 |
| | Dogs and Cats | 0.987 | 0.916 | 0.915 | 0.936 | 0.917 |
| | Imagenette | 0.989 | 0.958 | 0.908 | 0.916 | 0.93 |
| | Flowers 102 | 0.794 | 0.957 | 0.757 | 0.858 | 0.732 |
| | Oxford IIIT Pet | 0.93 | 0.932 | 0.884 | 0.911 | 0.889 |

# E  THE EFFECT OF CONFIDENCE LEVEL ON THE EFFICIENCY

Altering the confidence level enables users to balance explanation conciseness against reliability, as higher confidence levels ensure stronger fidelity guarantees, while lower levels yield more concise but less faithful explanations. The results, shown in Table 3 below, indicate that super-pixel–based functions consistently align better with the specified confidence levels than pixel-based ones. Therefore, super-pixel–based conformity functions are more reliable in upholding validity guarantees.

Table 3: The informational efficiency of FastSHAP explanations at different confidence levels using proposed conformity measures.

| Conformity Function | Dataset | 99% | 95% | 90% | 85% |
|---|---|---|---|---|---|
| Super-Pixels | Animals 10 | $0.49 \pm 0.165$ | $0.244 \pm 0.1$ | $0.175 \pm 0.075$ | $0.142 \pm 0.062$ |
| | Cards | $0.899 \pm 0.08$ | $0.658 \pm 0.126$ | $0.486 \pm 0.135$ | $0.382 \pm 0.127$ |
| | Dogs and Cats | $0.735 \pm 0.139$ | $0.361 \pm 0.134$ | $0.254 \pm 0.117$ | $0.2 \pm 0.104$ |
| | Imagenette | $0.588 \pm 0.187$ | $0.367 \pm 0.181$ | $0.271 \pm 0.157$ | $0.226 \pm 0.141$ |
| | Flowers 102 | $0.988 \pm 0.047$ | $0.948 \pm 0.122$ | $0.865 \pm 0.199$ | $0.775 \pm 0.25$ |
| | Oxford IIIT Pet | $0.882 \pm 0.113$ | $0.566 \pm 0.146$ | $0.45 \pm 0.136$ | $0.374 \pm 0.128$ |
| Scaled Values | Animals 10 | $0.516 \pm 0.18$ | $0.268 \pm 0.123$ | $0.186 \pm 0.089$ | $0.149 \pm 0.071$ |
| | Cards | $0.855 \pm 0.088$ | $0.57 \pm 0.155$ | $0.415 \pm 0.156$ | $0.33 \pm 0.136$ |
| | Dogs and Cats | $0.447 \pm 0.141$ | $0.283 \pm 0.11$ | $0.216 \pm 0.099$ | $0.172 \pm 0.089$ |
| | Imagenette | $0.452 \pm 0.192$ | $0.337 \pm 0.168$ | $0.248 \pm 0.136$ | $0.202 \pm 0.117$ |
| | Flowers 102 | $0.99 \pm 0.053$ | $0.9 \pm 0.189$ | $0.819 \pm 0.257$ | $0.762 \pm 0.289$ |
| | Oxford IIIT Pet | $0.82 \pm 0.128$ | $0.547 \pm 0.145$ | $0.434 \pm 0.13$ | $0.373 \pm 0.119$ |
| Pixelwise | Animals 10 | $0.834 \pm 0.055$ | $0.302 \pm 0.079$ | $0.166 \pm 0.051$ | $0.118 \pm 0.038$ |
| | Cards | $0.99 \pm 0.006$ | $0.913 \pm 0.029$ | $0.642 \pm 0.069$ | $0.416 \pm 0.05$ |
| | Dogs and Cats | $0.467 \pm 0.14$ | $0.189 \pm 0.079$ | $0.114 \pm 0.058$ | $0.083 \pm 0.047$ |
| | Imagenette | $0.802 \pm 0.171$ | $0.374 \pm 0.148$ | $0.161 \pm 0.072$ | $0.108 \pm 0.055$ |
| | Flowers 102 | $0.999 \pm 0.002$ | $0.996 \pm 0.006$ | $0.993 \pm 0.009$ | $0.99 \pm 0.013$ |
| | Oxford IIIT Pet | $0.978 \pm 0.015$ | $0.823 \pm 0.083$ | $0.662 \pm 0.145$ | $0.528 \pm 0.157$ |
| Summed Values | Animals 10 | $0.947 \pm 0.182$ | $0.739 \pm 0.39$ | $0.586 \pm 0.436$ | $0.454 \pm 0.436$ |
| | Cards | $0.924 \pm 0.219$ | $0.763 \pm 0.361$ | $0.633 \pm 0.408$ | $0.501 \pm 0.41$ |
| | Dogs and Cats | $0.6 \pm 0.419$ | $0.409 \pm 0.419$ | $0.315 \pm 0.393$ | $0.257 \pm 0.366$ |
| | Imagenette | $0.972 \pm 0.127$ | $0.782 \pm 0.384$ | $0.613 \pm 0.456$ | $0.529 \pm 0.467$ |
| | Flowers 102 | $0.99 \pm 0.0$ | $0.99 \pm 0.0$ | $0.99 \pm 0.0$ | $0.99 \pm 0.0$ |
| | Oxford IIIT Pet | $0.969 \pm 0.131$ | $0.91 \pm 0.244$ | $0.854 \pm 0.308$ | $0.784 \pm 0.363$ |

# F  THE EFFECT OF CONFIDENCE LEVEL ON THE FIDELITY

The super-pixel–based functions yield fidelity levels that align more closely with the predefined confidence levels than the pixel-wise function. Consequently, they provide greater reliability in upholding the validity guarantees.

Table 4: The fidelity of FastSHAP explanations at different confidence levels using proposed conformity measures.

| Conformity Function | Dataset | 99% | 95% | 90% | 85% |
|---|---|---|---|---|---|
| Super-Pixels | Animals 10 | 0.974 | 0.93 | 0.882 | 0.844 |
| | Cards | 0.985 | 0.97 | 0.936 | 0.857 |
| | Dogs and Cats | 0.981 | 0.949 | 0.92 | 0.903 |
| | Imagenette | 0.988 | 0.96 | 0.929 | 0.901 |
| | Flowers 102 | 0.98 | 0.927 | 0.861 | 0.8 |
| | Oxford IIIT Pet | 0.969 | 0.905 | 0.863 | 0.814 |
| Scaled Values | Animals 10 | 0.98 | 0.941 | 0.891 | 0.854 |
| | Cards | 0.977 | 0.966 | 0.909 | 0.845 |
| | Dogs and Cats | 0.976 | 0.943 | 0.914 | 0.891 |
| | Imagenette | 0.989 | 0.965 | 0.934 | 0.91 |
| | Flowers 102 | 0.981 | 0.913 | 0.853 | 0.817 |
| | Oxford IIIT Pet | 0.964 | 0.912 | 0.866 | 0.834 |
| Pixelwise | Animals 10 | 0.952 | 0.869 | 0.798 | 0.743 |
| | Cards | 0.989 | 0.936 | 0.811 | 0.736 |
| | Dogs and Cats | 0.956 | 0.887 | 0.841 | 0.812 |
| | Imagenette | 0.964 | 0.848 | 0.807 | 0.777 |
| | Flowers 102 | 0.995 | 0.981 | 0.974 | 0.968 |
| | Oxford IIIT Pet | 0.968 | 0.876 | 0.777 | 0.705 |
| Summed Values | Animals 10 | 0.98 | 0.924 | 0.872 | 0.814 |
| | Cards | 0.974 | 0.94 | 0.902 | 0.838 |
| | Dogs and Cats | 0.962 | 0.916 | 0.876 | 0.839 |
| | Imagenette | 0.993 | 0.958 | 0.909 | 0.876 |
| | Flowers 102 | 0.956 | 0.956 | 0.956 | 0.956 |
| | Oxford IIIT Pet | 0.97 | 0.932 | 0.894 | 0.842 |

# G   PLOTS FOR THE EFFECT OF CONFIDENCE LEVEL ON THE EFFICIENCY AND THE FIDELITY

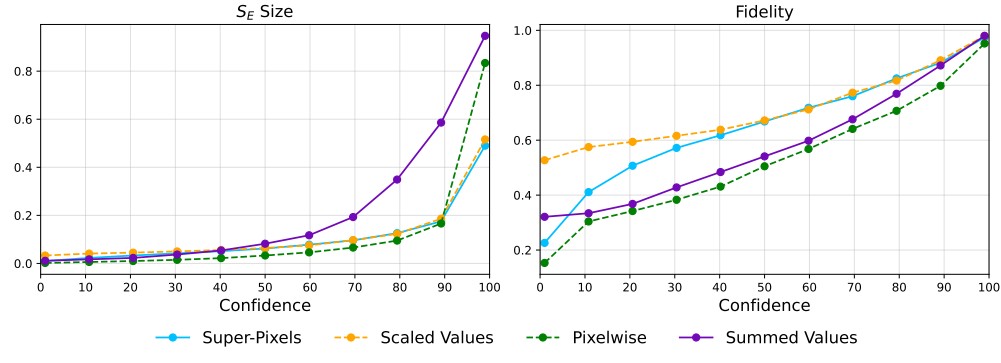

(a) Animals Dataset

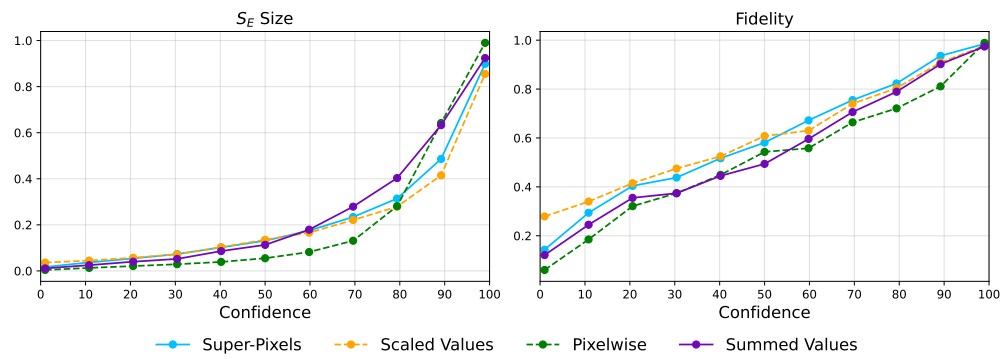

(b) Cards Dataset

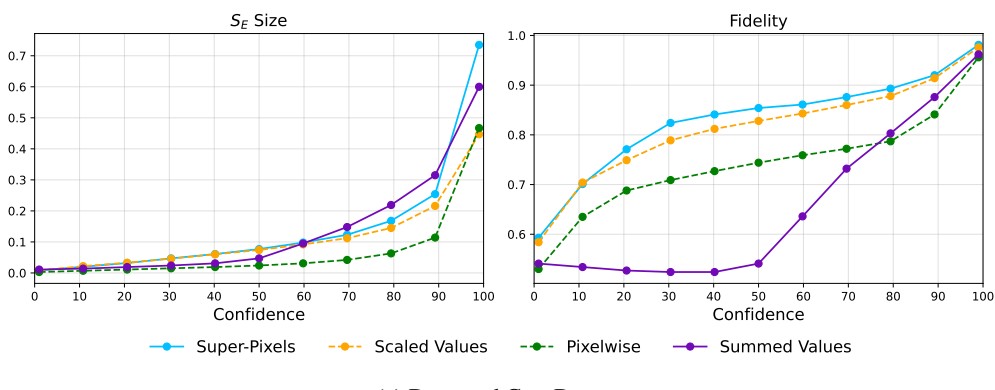

(c) Dogs and Cats Dataset

Figure 8: The effect of the confidence level on the size of $S_E$ and the fidelity level of the retrieved explanations using FastSHAP with the four proposed conformity functions.

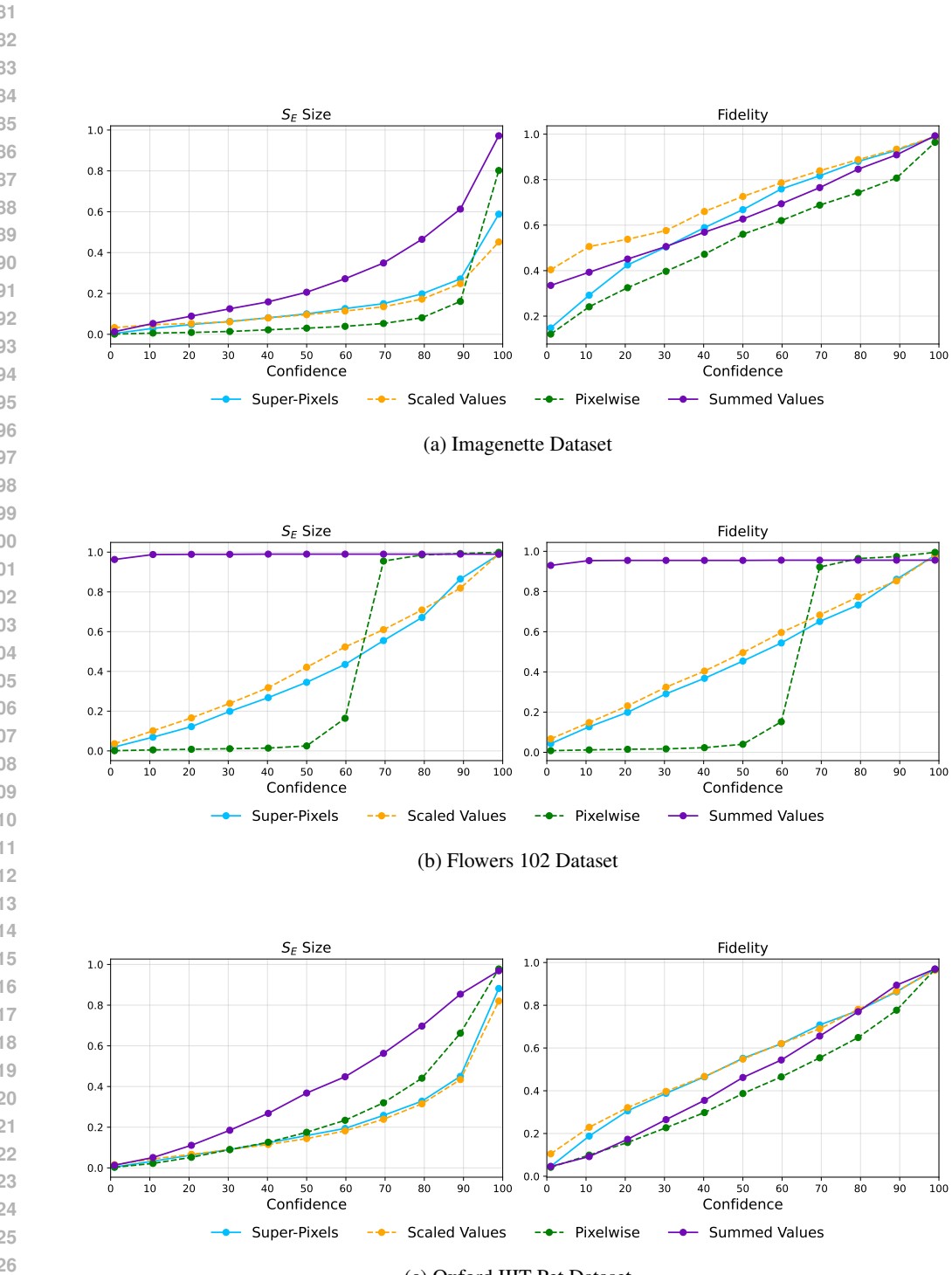

(a) Imagenette Dataset

(b) Flowers 102 Dataset

(c) Oxford IIIT Pet Dataset

Figure 9: The effect of the confidence level on the size of $S_E$ and the fidelity level of the retrieved explanations using FastSHAP with the four proposed conformity functions.

# H DATASET DETAILS AND HARDWARE SPECIFICATIONS

The experiments were conducted in a Python environment on a system equipped with an Intel(R) Core(TM) Ultra 9 185H CPU (2.30 GHz) and 64 GB of RAM, with GPU support from an NVIDIA® GeForce® RTX 4070. Additional experiments were carried out using an NVIDIA Tesla V100f GPU and Intel(R) Xeon(R) Gold 6226R CPU @ 2.90GHz.

Table 5 presents an overview of the datasets used in the experiments. The table includes the number of classes, number of features, dataset size, training, validation, and test split sizes. Additionally, the table provides the corresponding dataset ID from OpenML.

Table 5: The dataset information.

| Dataset | # Classes | Dataset Size | Train. Set | Val. Set | Test Set | Cal. Set | URL |
|---|---|---|---|---|---|---|---|
| Animals 10 | 10 | 26,179 | 19,895 | 1,048 | 3,900 | 1,336 | https://kaggle.com/datasets/alessiocorrado99/animals10 |
| Cards | 53 | 8,154 | 6,099 | 265 | 265 | 1,525 | https://tinyurl.com/mry4tdk7 |
| Dogs and Cats | 2 | 37,500 | 23,500 | 1,500 | 9,312 | 3,188 | https://kaggle.com/competitions/dogs-vs-cats |
| Imagenette | 10 | 13,394 | 8,522 | 947 | 1,001 | 2,924 | https://github.com/fastai/imagenette |
| Flowers 102 | 102 | 8,189 | 1,020 | 1,020 | 4,581 | 1,568 | https://robots.ox.ac.uk/˜vgg/data/flowers/102/ |
| Oxford IIIT Pet | 37 | 7,349 | 3,312 | 368 | 2,733 | 936 | https://robots.ox.ac.uk/˜vgg/data/pets/ |

# I  PSEUDO-CODE

**Algorithm A Python Pseudo-Code for Calibration:** We provide code from our Python implementation of the calibration using the Superpixels measure. Specifically, the implementation includes superpixel extraction, functions for selecting important superpixels, and the calibration step.

```python
import torch
import numpy as np
from skimage.segmentation import slic

def extract_superpixels(images, n_segments=50):
    """
    Extract superpixels from a batch of images using the SLIC algorithm.
    """
    img_segmentations = []
    for img in images:
        # Rearrange tensor dimensions from [C, H, W] to [H, W, C]
        img = img.permute(1, 2, 0).numpy()
        # Apply the SLIC algorithm to divide the image into superpixels
        segments_ = slic(img, n_segments=n_segments, compactness=10, sigma=1, start_label=1)
        img_segmentations.append(segments_)
    return img_segmentations

def extract_masks(attributions):
    """
    Generate binary masks from feature attributions by thresholding at different percentiles.
    """
    attributions_flat = attributions.reshape(attributions.shape[0], -1)
    # Stores binary masks at different thresholds
    importance_mask_dict = {}
    # Stores the numerical thresholds for each mask
    mask_thr_dict = {}
    for p in reversed(range(1, 100)):# keeping the top 1%, 2%, ..., 99% of features
        thresholds = np.percentile(attributions_flat, p, axis=1)
        importance_mask_dict[str(100-p)] = np.array(
            [val >= tr for val, tr in zip(attributions, thresholds)]
        ).astype(int)
        mask_thr_dict[str(100-p)] = np.array([tr for tr in thresholds]).astype(float)
    return importance_mask_dict, mask_thr_dict

def extract_important_superpixels(img_segmentations, importance_masks, rho):
    """
    Select important superpixels
    """
    seg_masks = []
    for s in range(len(img_segmentations)):
        segmentation = img_segmentations[s]
        mask = importance_masks[s].squeeze().numpy()
        seg_flat = segmentation.flatten()
        mask_flat = mask.flatten()
        overlap_per_seg = np.bincount(seg_flat, weights=mask_flat)
        important_seg_ids = np.where(overlap_per_seg >= rho)[0]
        seg_mask_ = np.isin(segmentation, important_seg_ids).astype(np.uint8)
        seg_masks.append(np.expand_dims(seg_mask_, axis=0))
    return seg_masks

def calibrate(blackbox_model, cal_data, cal_preds, img_segmentations, importance_mask_dict,
              mask_thr_dict, epsilone=5, rho=100):
    """
    Compute the conformity score at the predefined significance level
    """
    prediction_thresholds = []
    for p, importance_masks in importance_mask_dict.items():
        # Thresholds for the current percentile
        thresholds = mask_thr_dict[p]
        importance_masks = torch.tensor(importance_masks, dtype=torch.float32).unsqueeze(1)
        seg_masks = extract_important_superpixels(img_segmentations, importance_masks, rho)
        masked_images = cal_data * np.array(seg_masks)
        mask_preds = torch.max(blackbox_model(masked_images.float()), dim=-1)[1]
        # Boolean vector indicating which masked predictions are correct
        correct = cal_preds == mask_preds
        # Store thresholds only for correctly predicted images
        prediction_thresholds.append(torch.from_numpy(thresholds) * correct)
        # If all predictions are correct, stop
        if torch.all(correct).item():
            break
    stacked_thresholds = torch.stack(prediction_thresholds)
    # Replace zeros of incorrect predictions with -inf so they don't affect max operation
    stacked_thresholds[stacked_thresholds == 0] = float('-inf')
    max_val = torch.max(stacked_thresholds, dim=0)[0]
    # Compute epsilon-percentile calibration threshold across samples
    sigma_ = np.percentile(max_val.detach().cpu().numpy(), epsilone)
    return sigma_
```

**Algorithm B Python Pseudo-Code for Inference:** we provide code from our Python implementation for extracting the sufficient explanation regions ($S_E$) during inference.

```python
import numpy as np

def extract_sufficient_explanations(images_, attributions, sigma_, img_segmentations, rho=100):
    """
    Extract a sufficient explanation for an image by masking out
    unimportant regions based on feature attributions and superpixel segmentation.

    Args:
        images_ (torch.Tensor): Batch of input images, shape [N, C, H, W].
        attributions (np.ndarray or torch.Tensor): Attribution values for each pixel,
            shape [N, H, W] or [N, C, H, W].
        sigma_ (float): Conformity score (threshold) at the predefined confidence level.
                        Pixels with attribution >= sigma_ are considered important.
        img_segmentations (list of np.ndarray): Superpixel segmentation maps for each image,
            each with shape [H, W], where values are superpixel IDs.
        rho (int): Minimum number of important pixels required for a superpixel
            to be considered important as a whole.

    Returns:
        torch.Tensor: Masked images containing only the sufficient explanation,
                      shape [N, C, H, W].
    """
    attributions = np.array(attributions)
    # Create pixel-level importance masks:
    # 1 if attribution >= sigma_ (important), 0 otherwise
    importance_masks = np.array([val >= sigma_ for val in attributions]).astype(int)
    # keep only superpixels that contain at least `rho` important pixels
    seg_masks = extract_important_superpixels(img_segmentations, importance_masks, rho)
    # Apply the refined masks to the images to retain only important regions
    sufficient_explanation = images_ * seg_masks
    # Return the explanation regions as torch tensors
    return sufficient_explanation.float()
```

