# OpenReview forum: "Learning Conformal Explainers for Image Classifiers"
_ICLR.cc/2026/Conference — ICLR 2026 Conference Withdrawn Submission_

### Official Review · Reviewer_yaRq · 2025-10-23

**Soundness:** 3
**Presentation:** 2
**Contribution:** 2
**Rating:** 4
**Confidence:** 3

**Summary:**

This paper investigates the explainability problem of image classifiers. Building on existing feature attribution methods, the authors propose an approach that determines the minimal sufficient feature subset to maintain the original model prediction. This minimal subset provides a qualitative aspect that complements the quantitative assessments on feature influences provided by feature attribution methods. In addition to the proposed algorithm, the authors further introduce variants of the conformity function, which exhibit different behaviors in terms of confidence and fidelity during empirical evaluation.

**Strengths:**

- This work provides a new perspective for presenting the results of feature attributions, which is generally orthogonal to existing explainability research.
- The main results are supported by detailed quantitative evaluations presented in the appendix.
- The provided pseudocode facilitates a clearer understanding of the proposed algorithm and enhances the reproducibility of the presented results.

**Weaknesses:**

- It is unclear how this work should be positioned in the established explainability literature. I personally interpret it as a plugin that complements existing feature attribution methods, but its connection and contribution should be better clarified and communicated.
- The motivation and the reported results are somewhat disconnected. This work begins with a critique of the robustness and faithfulness of feature attribution methods; however, it remains unclear how the conformal explainer addresses these limitations. In fact, the proposed method will inherit, if any, the weaknesses of the underlying explainer that produces $\Phi$, given its direct reliance on feature attributions.
- The writing and overall presentation of this work represent another major weakness. Many notations collected from different sources are presented together without alignment. Some discussions and even entire sections appear contextually unfit, missing the necessary motivation or clarification for the particular designs.
- Although the experimental results are detailed, they only focus on comparisons among the variants within the conformal explainer framework. The results do not show how the proposed approach contributes beyond the existing literature, which limits its potential impact.

**Questions:**

- The extended discussion on marginal and conditional expectations is a bit confusing. These are neither new proposals nor essential for the following discussion. It appears that the preliminary section would be a better fit to include general information about available baseline choices.
- Both sections 3.3 and 3.4 read as rather isolated. What is the purpose, motivation, or intuition behind the approaches described there?
- As mentioned in the weakness part, the symbol system is not well aligned. Related to this point:
  - The symbols $y_i$, $h(\cdot)$, and $f(\cdot)$ all refer to model outcomes but are represented differently.
  - The symbol $f_i$ (line 199) denotes an input feature but conflicts with the use of $f(\cdot)$ (line 184). Is there any inherent connection between them?
  - The calibration dataset $\mathbb{Z}$ refers to input-prediction tuples (Section 2.1, line 90) but is redefined as a set of input-attribution tuples, where $\Phi$ represents attribution vectors.
- Why are some plots dashed and others solid?

---

> ### Author Response · Authors · 2025-11-13
>
> We sincerely thank the reviewer for their careful and constructive feedback.
>
> >1- It is unclear how this work should be positioned in the established explainability literature. I personally interpret it as a plugin that complements existing feature attribution methods, but its connection and contribution should be better clarified and communicated.
>
> Our work is indeed designed as a post-hoc processing, or “plugin,” as the reviewer aptly put it, that builds on existing feature attribution methods. The key contribution lies not in generating attributions, but in assigning each new explanation a confidence score (a fidelity level) without going through expensive operations. The calibration produces formally valid and interpretable sufficient explanations that are guaranteed to preserve essential features for the predictions with user-controlled confidence.
>
> The proposed approach is orthogonal to conventional attribution methods: rather than competing with them, it provides a formal layer of reliability that can be applied across methods. We will explicitly clarify this positioning in the introduction and discussion sections.
>
> >2- In fact, the proposed method will inherit, if any, the weaknesses of the underlying explainer that produces, given its direct reliance on feature attributions.
>
> Actually, the proposed method is designed to reveal the uncertainty and limitations of the underlying explainer. For example, if the explainer produces the entire image as an explanation at high confidence levels, this behavior indicates substantial uncertainty and randomness, suggesting that the explainer’s outputs are unreliable and should not be trusted.
>
> >3- On the clarity and alignment of notation
>
> We thank the reviewer for highlighting the inconsistencies in notation. We acknowledge this as an issue that must be addressed and will revise the paper to ensure coherence.
>
> >4- The extended discussion on marginal and conditional expectations is a bit confusing. These are neither new proposals nor essential for the following discussion. It appears that the preliminary section would be a better fit to include general information about available baseline choices.
>
> We totally agree with the reviewer, and further clarification will be carried out.
>
> >5- Both sections 3.3 and 3.4 read as rather isolated. What is the purpose, motivation, or intuition behind the approaches described there?
>
> Sections 3.3 and 3.4 introduce alternative conformity functions, “Scaled Values” and “Summed Values”, designed to explore how different normalization or accumulation schemes affect the trade-off between explanation compactness and predictive fidelity.
>
> Their purpose is to test the generality and flexibility of the conformal calibration principle under different conformity functions.
>
> >6- The symbol $f_i$ (line 199) denotes an input feature but conflicts with the use of $f(\cdot)$ (line 184). Is there any inherent connection between them?
>
> There is no inherent connection between them. $f(\cdot)$ must be replaced by $h(\cdot)$.
>
> >7- The calibration dataset $\mathbb{Z}$ refers to input-prediction tuples (Section 2.1, line 90) but is redefined as a set of input-attribution tuples, where $\Phi$ represents attribution vectors.
>
> In the conventional formulation of conformal prediction, $\mathbb{Z}$ represents input–target tuples, as it requires a label to compute the non-conformity score. In the case of our proposed method, the conformity function operates on the entire vector of attribution scores to compute the conformity score.
>
> >8- Why are some plots dashed and others solid?
>
> The motivation is to improve the visualization of regions with a high concentration of plot lines.

---

> > ### Comment · Reviewer_yaRq · 2025-11-20
> >
> > I appreciate the authors’ detailed responses. I believe the paper would benefit from incorporating the clarifications provided during rebuttal, which could better guide readers more effectively through the design choices.
> >
> > That said, I will maintain my original score given the current version of the paper.

---

> > > ### Author Response · Authors · 2025-11-20
> > >
> > > We thank the reviewer for their comments. We have updated the paper according to their previous feedback and uploaded the revised version.

---

### Official Review · Reviewer_KDip · 2025-10-31

**Soundness:** 2
**Presentation:** 2
**Contribution:** 2
**Rating:** 2
**Confidence:** 5

**Summary:**

The paper proposed a metric to measure and control the degree of explanation for  post-hoc explainable AI (xAI) methods that measures the importance of attributions via the conformal prediction score, which basically reflects the variance of the classification probability under specified perturbation.

**Strengths:**

Compared with traditional Shapley value approaches which is in similar framework, the proposed method provide more statistical plausible measurement that can compare the degree of explanability over different situations.  Also, the method can be applied on different levels of features from pixel to super-pixel of vectors, which is more flexible.

**Weaknesses:**

Although the authors claim their method can be applied to all xAI models that generates feature attributions. The formulation of xAI models in Sect. 2.2 only fits primitive explanation models (perturbation or simple gradient methods) , but cannot be adopted to more advanced approaches such as class activation mapping, counter-factual generation or latent space disentanglement, which are more popular now.  The baseline explanation model they used in the experiments all fall in to the category of perturbation or simple gradient ones, and most of them are quite out-dated.
On the other hand, the experiments in Fig. 3 and 4 showed confidence vs. SE and confidence vs. fidelity, both are positively correlated. It doesn't justifies how the proposed measurement help with trading-off different critical characters of a baseline method, so it cannot persuade that the measurement is useful.

**Questions:**

The exact calculation of "fidelity" is completely missing in the manuscript. How did you define and calculate it? Also, between "fidelity" and "confidence", which one exactly defines "compactness"? As both are positively correlated but none of them are directly related to compactness (i.e.,  low dimensionality).

---

> ### Author Response · Authors · 2025-11-13
>
> We thank the reviewer for the feedback and insightful observations.
>
> >1- The formulation of xAI models in Sect. 2.2 only fits primitive explanation models (perturbation or simple gradient methods) , but cannot be adopted to more advanced approaches such as class activation mapping, counter-factual generation or latent space disentanglement, which are more popular now.
>
> The formulation in Section. 2.2 is laid as a brief example of a feature attribution method and is not meant to be an exhaustive list of all the existing feature attribution methods.
>
> Our method indeed builds on the feature attribution paradigm, as it requires an attribution vector $\Phi$ that quantifies the contribution of each feature, which can be obtained using any method that user decides on. While the examples in Section 4 employ gradient and perturbation-based methods, the proposed conformal calibration framework is agnostic to the underlying explanation mechanism, as long as feature-level importance values are available.
> Therefore, the proposed approach is orthogonal to conventional attribution methods and does not compete with them.
>
> >2- The baseline explanation model they used in the experiments all fall in to the category of perturbation or simple gradient ones, and most of them are quite out-dated.
>
> The reviewer is correct that the chosen explainers (e.g., GradientSHAP, KernelSHAP, and FastSHAP) primarily belong to the family of perturbation and gradient-based methods. Our rationale for this selection is twofold:
>
> 1- These methods represent the most widely used and well-understood attribution families in the literature, allowing controlled comparison across different conformity functions.
>
> 2- Several recent works (e.g., [1] and [2]) continue to use these methods as standard benchmarks for evaluating new explanation paradigms, ensuring fair and reproducible results.
>
> Again, nothing prevents the user from using any method as long as it provides feature-level importance scores.
>
> >3-  the experiments in Fig. 3 and 4 showed confidence vs. SE and confidence vs. fidelity, both are positively correlated. It doesn't justifies how the proposed measurement help with trading-off different critical characters of a baseline method, so it cannot persuade that the measurement is useful.
>
> The reviewer is correct that both confidence and fidelity are positively correlated. However, this observation is indeed expected and consistent with the conformal prediction framework. The value of the proposed conformity-based measure lies in allowing users to control and quantify this trade-off with formal guarantees, something that traditional attribution-based explainers do not provide. By adjusting the significance level $\epsilon$, the user can systematically balance interpretability (compactness) against predictive fidelity. This property is analogous to the efficiency–validity trade-off in standard conformal prediction ([3] and [4]) and represents a key contribution of our work.
>
> To further clarify:
>
> $\textbf{Confidence ($1 − \epsilon$):}$  is the user-specified calibration level that defines the probability of retaining all features necessary to preserve the model’s decision.
>
> $\textbf{Fidelity:}$ quantifies the empirical agreement between the model’s prediction on the sufficient explanation ($S_E$) and the full input. Formally, fidelity is computed as the ratio between the number of test instances for which both predictions coincide and the total number of test instances. This definition is available in lines 298 and 299 (" the fidelity of the generated explanations, i.e., the degree to which the explanations preserve the same prediction of the model when provided with the full input image.") will be further clarified and added explicitly to Section 4.3 for clarity.
>
> $\textbf{Compactness (informational efficiency):}$ is measured by the relative size of the SE region, i.e., the proportion of retained features or superpixels. As confidence increases, the $S_E$ becomes larger (less compact) to preserve higher fidelity. Thus, the framework enables an explicit and quantifiable trade-off between fidelity (predictive faithfulness) and compactness (explanation succinctness).
>
>
>
> [1]- Jethani, N., Sudarshan, M., Covert, I. C., Lee, S.-I., and Ranganath, R. FastSHAP: Real-time shapley value estimation. In ICLR, 2022.
>
> [2]- Pan, D., Moniz, N., and Chawla, N. V. Fast explanations via policy gradient-optimized explainer. In Proceedings of IJCAI-25.
>
> [3]- H. Linusson, U. Johansson, and H. Boström. Efficient conformal predictor ensembles. Neurocomputing, 397:266–278, 2020.
>
> [4]- Glenn Shafer and Vladimir Vovk. A tutorial on conformal prediction. J. Mach. Learn. Res., 9:371–421, June 2008. ISSN 1532-4435.

---

> > ### Comment · Reviewer_KDip · 2025-11-22
> >
> > I thank the authors for their detailed clarification and I believe the method can be extended to more complex approaches. However, I don't think that would be a plain work and more supporting evidences need to be provided. Hence I maintain the initial score.

---

> > > ### Author Response · Authors · 2025-11-23
> > >
> > > We agree with the reviewer that the proposed approach could be extended to more complex methods. We would be grateful if the reviewer could clarify what additional supporting evidence they believe is needed. Could they please specify the type of evidence they find lacking?

---

> > > > ### Comment · Reviewer_KDip · 2025-11-26
> > > >
> > > > In my opinion, experimental validations on Grad-CAM and a counter-factual generation method (such as ICAM-reg) should be necessary.

---

### Official Review · Reviewer_XjgV · 2025-11-01

**Soundness:** 3
**Presentation:** 2
**Contribution:** 2
**Rating:** 2
**Confidence:** 4

**Summary:**

Brief summary: This paper proposes an approach to calibrate a global threshold based on existing feature attribution approaches, so that keeping the highest attributed features preserves the confidence for a predicted class.

**Strengths:**

- This paper borrows the concept of conformal prediction and applies it in the context of feature attribution. The developed method determines a global threshold over attributions derived from existing explainers, which enables a qualitative presentation of explanation results by showing the minimal components that preserve the original prediction.

**Weaknesses:**

- While the authors claim the development of a method that “generates explanations”, the conformal explainer leans towards a post-hoc processing of the explanations, as it relies on existing information and does not determine attributions itself.
- The symbol system lacks internal consistency, making it sometimes difficult to associate the abstractions with their corresponding concepts and to follow the description of the method. Particularly, the use of $f_j$ to denote features is very confusing, as the same is used, and commonly used, to denote functions in lines 174~186.
- While conformal prediction is well-defined with model predictions $y_i$ as described by Eq. (1), there is no inherent guarantee in a conformal explainer that translates the threshold $\tau$ for attributions to the prediction outcome space. The authors enforce label preservation in Eq. (3) through an explicit constraint; yet, the quality of the presented subsets is still dominated by the underlying feature attribution methods.
- The time costs are not reported, which could be another concern given the repeated queries required to determine $\sigma_i$ as specified in Eq. (3).

**Questions:**

- Could the authors elaborate further on the motivation for identifying a global threshold for subset presentation? The tested feature attribution methods all provide local explanations, with patterns that can vary across different test cases (depending on the analyzed classes and the local decision boundaries at a particular location of the manifold ). Given the “local” nature of those explanations, wouldn’t determining a specific threshold for each explanation be more effective and also easier?

---

> ### Author Response · Authors · 2025-11-13
>
> We thank the reviewer for their careful reading of our submission and for providing valuable and constructive feedback.
>
>
> >1- Could the authors elaborate further on the motivation for identifying a global threshold for subset presentation? The tested feature attribution methods all provide local explanations, with patterns that can vary across different test cases (depending on the analyzed classes and the local decision boundaries at a particular location of the manifold ). Given the “local” nature of those explanations, wouldn’t determining a specific threshold for each explanation be more effective and also easier?
>
> Our primary objective is to enable users to assign a controllable level of confidence (fidelity) to the generated explanations, independent of the specific explanation method or its inherent accuracy. Furthermore, the principles of conformal prediction provide formal validity and ensure comparability across instances. In particular, by employing a single global threshold, we can guarantee, under the probabilistic exchangeability assumption, that the sufficient explanation set contains all features necessary to the prediction with a user-defined confidence level of $1 - \epsilon$, which is achieved without incurring the prohibitive computational cost of repeatedly evaluating the underlying black-box model multiple times for each new observation until $\sigma_i$​ is determined. In addition, the size of the regions derived through conformal prediction reflects the degree of uncertainty, thereby facilitating the comparison of different explanation methods.
>
> We definitely agree that patterns may vary across different test cases. However, once a particular pattern is recognized by the explainer, it assigns corresponding importance scores to its associated features. Our proposed approach operates at the level of the importance scores, identifying the magnitude of importance that the explainer attributes to the relevant patterns.
>
> >2- While the authors claim the development of a method that “generates explanations”, the conformal explainer leans towards a post-hoc processing of the explanations, as it relies on existing information and does not determine attributions itself.
>
> We agree with the reviewer that the proposed approach is mainly a post-hoc processing, and that the term “generates” does not accurately describe the method. We will revise the manuscript accordingly to ensure that the term is used with consistency.
>
> >3- The symbol system lacks internal consistency
>
> We thank the reviewer for pointing out the inconsistency in the notation. We acknowledge this as an oversight and will revise the notation to ensure internal consistency.

---

> ### Author Response · Authors · 2025-11-13
>
> >4- While conformal prediction is well-defined with model predictions $y_i$ as described by Eq. (1), there is no inherent guarantee in a conformal explainer that translates the threshold $\tau$ for attributions to the prediction outcome space. The authors enforce label preservation in Eq. (3) through an explicit constraint; yet, the quality of the presented subsets is still dominated by the underlying feature attribution methods.
>
> We would like to clarify that our approach does not claim to generate raw attributions, rather, it transforms existing attributions into statistically valid sufficient explanations. Such a transformation is non-trivial: it calibrates the feature importance scores such that the resulting subsets are guaranteed, at a specified confidence level, to reproduce the model’s prediction. In this sense, the method provides an additional explanatory layer that quantifies uncertainty and enforces predictive fidelity, extending the utility of existing explainers rather than replacing them.
>
> We agree that the threshold $\sigma_\epsilon$ operates in the attribution space and not directly in the model’s output space. However, this is precisely why we enforce the label preservation constraint in Eq. (3), ensuring that the selected subset of features yields the same prediction as the full input. The validity guarantee of Theorem 1 formalizes this connection, bounding the probability of omitting a feature critical for preserving the model’s prediction by $\epsilon$.
>
> >5- The time costs are not reported, which could be another concern given the repeated queries required to determine $\sigma_i$ as specified in Eq. (3).
>
> That is absolutely a fair point. We will report the computational complexity in the paper for both calibration and inference. Here, I will report the complexity of the function in Eq. 4, which is the most efficient and most computationally demanding:
>
> First, the complexity of segmenting all the images is $\mathcal{O}(N \times M)$ as we emply SLIC algorithm which is linear in the number of pixels ($M$) in the image $\mathcal{O}(M)$ [1]
>
> The computational complexity of the calibration procedure can be summarized as follows. The overall cost is dominated by the repeated model evaluations across multiple importance thresholds. Specifically, for a calibration set of $N$ images of size ($M = H \times W$), cost of model evaluation $\Omega$, and $P$ percentile thresholds, the computational complexity is:
>
> $\mathcal{O}(P \times (N \times M + N \times \Omega))$.
>
> The first term accounts for superpixel extraction, mask generation, and thresholding operations, each linear in the number of pixels, while the second term represents the repeated forward passes of the black-box model on masked images. Since the model inference cost typically dominates, the total complexity can be approximated as
>
> $\mathcal{O}(P \times N \times \Omega)$.
>
>
> The computational complexity at inference time is primarily linear in the number of pixels per image. The function first computes pixel-level importance masks through a thresholding operation using $\sigma_\epsilon$ with cost $\mathcal{O}(M)$. Then we aggregate pixel-wise importance values over superpixels and applies a filtering criterion, also with linear complexity $\mathcal{O}(M)$. The subsequent element-wise masking of the images incurs another $\mathcal{O}(C \times M)$ where $C$ is the number of channels in the input image. Consequently, the overall time complexity is
> $\mathcal{O}(C \times M)$.
> In practice, the computational cost is dominated by simple array operations and scales linearly with image size, making the procedure highly efficient.
>
>
> [1] - Radhakrishna Achanta, Appu Shaji, Kevin Smith, Aurelien Lucchi, Pascal Fua, and Sabine Susstrunk. Slic superpixels compared to state-of-the-art superpixel methods. IEEE Transactions on Pattern Analysis and Machine Intelligence, 34(11):2274–2282, 2012.

---

### Author Response · Authors · 2025-11-13

We sincerely thank all reviewers for their time and thoughtful evaluation of our submission. We greatly appreciate the constructive feedback and the valuable comments that will help us improve the quality and clarity of our work.

Although we recognize that the paper may not have a meaningful opportunity for acceptance given the current scores, we decided to carefully address all the questions and comments raised by the reviewers out of respect for the effort they devoted to reviewing our work and providing helpful insights.

---

### Author Response · Authors · 2025-11-20

Dear Reviewers,

Thank you for your constructive feedback. We have revised the manuscript in accordance with your comments and have uploaded an updated version of the paper. We remain fully committed to improving the quality of our work and are willing to make any further revisions based on your additional comments or suggestions.

Best regards

---

### Note · Authors · 2026-01-26

I have read and agree with the venue's withdrawal policy on behalf of myself and my co-authors.

---

### Meta-Review · Area_Chair_Qym2 · 2026-01-06

**Summary:**

Overall, I think this version—as all reviewers agreed—is not ready for publication at this conference. The main weakness lies in the lack of a clear and convincing contribution: although the authors claim to “generate explanations,” the conformal explainer mainly acts as a post-hoc wrapper that relies entirely on existing attribution methods and therefore inherits their limitations. The positioning of the work within the explainability literature is unclear, and the experiments do not demonstrate added value beyond prior approaches, especially since they focus only on outdated perturbation and simple gradient methods rather than modern xAI techniques. In addition, the experimental results fail to justify the usefulness of the proposed measurements, as the reported correlations do not show meaningful trade-offs. The paper also raises theoretical and practical concerns, including the absence of guarantees that conformal thresholds on attributions translate to prediction outcomes and the lack of analysis of computational cost. Finally, the overall presentation is weak: inconsistent notation, unclear motivation, and poorly aligned sections make the method difficult to follow and further limit the impact of the work.

**Reviewer Concerns:**

All reviewers agree that this version is not ready for publication, as the proposed conformal explainer offers limited novelty and mainly acts as a post-hoc wrapper that inherits the weaknesses of existing attribution methods. The paper’s contribution and positioning in the explainability literature are unclear, and the experiments—focused on outdated baselines—do not convincingly demonstrate added value or practical usefulness. In addition, theoretical guarantees, efficiency analysis, and clear presentation are all lacking, which further limits the impact of the work.

**Reviewer Scores:**

The authors attempted to address the concerns raised; however, they are not fully convincing, and the paper requires further work to meet the standards of this conference.

---

### Decision · Program_Chairs · 2026-01-26

Reject